# Adherence to Mediterranean Diet Is Associated with Multiple Sclerosis Severity

**DOI:** 10.3390/nu15184009

**Published:** 2023-09-16

**Authors:** Monica Guglielmetti, Wahidah H. Al-Qahtani, Cinzia Ferraris, Giuseppe Grosso, Simona Fiorini, Eleonora Tavazzi, Giacomo Greco, Alessandro La Malfa, Roberto Bergamaschi, Anna Tagliabue

**Affiliations:** 1Human Nutrition and Eating Disorder Research Center, Department of Public Health, Experimental and Forensic Medicine, University of Pavia, 27100 Pavia, Italy; monica.guglielmetti@unipv.it (M.G.);; 2Laboratory of Food Education and Sport Nutrition, Department of Public Health, Experimental and Forensic Medicine, University of Pavia, 27100 Pavia, Italy; 3Department of Food Sciences & Nutrition, College of Food & Agriculture Sciences, King Saud University, Riyadh 11451, Saudi Arabia; 4Department of Biomedical and Biotechnological Sciences, University of Catania, 95123 Catania, Italy; 5IRCCS Mondino Foundation, 27100 Pavia, Italy

**Keywords:** Mediterranean diet, multiple sclerosis, dietary habits, multiple sclerosis severity

## Abstract

Currently available data suggest that the union of a balanced diet and an overall healthy lifestyle may determine an amelioration in several clinical parameters and in the quality of life for patients with MS (pwMS). The study objective was to investigate the possible difference in MS severity in a group of Italian patients with MS based on their adherence to Mediterranean Diet (MedDiet). Eating habits were collected through a validated 110-items Food Frequency Questionnaire, the Medi-Lite score was used for adherence to MedDiet evaluation. MS severity was graded according to Herbert’s severity scale, based on the MSSS. 106 patients were classified in 3 groups according to their MedDiet adherence (low/medium/high). Higher adherence was associated with a 6.18 (95% CI: 1.44, 26.59) higher probability of having a mild-to-moderate MS. When studying the single constituents of the Medi-Lite score, none of them was individually associated with MS severity. It remains unclear whether effects of specific dietary components included in the MedDiet may impact the health status at disease onset or can slow down the symptoms due course of disease. Future studies are needed to reproduce our findings and should focus on answering the latter raised question.

## 1. Introduction

The Mediterranean Diet (MedDiet) was firstly studied by Keys in the 1960s and refers to the dietary and lifestyle habits of the populations of the Mediterranean sea during those years. At the basis of MedDiet principles there are seasonality, conviviality, adequate rest, regular physical activity and the preference for traditional and local foods [1]. Among these foods, fruit and vegetables, nuts, legumes, cereals (mainly whole grains) and fish should be preferred, milk and dairy products should be moderately consumed while meat and meat products should be limited. MedDiet traditionally encourages olive oil and moderate alcohol consumption (such as red wine) ideally during meals [1,2]. It has been recognized that MedDiet is a healthy diet with potential beneficial effects on several chronic diseases, and multiple sclerosis (MS) may be one of them [3]. MS is one of the primary determinants of disability in young adults [4] and it presents with a variety of symptoms, which can affect the motor, sensory, cognitive and visceral systems [5,6]. The exact pathological process through which MS develops and progresses is still not known. Certainly, it is a complex condition, in which immune [7], genetic [8], and environmental factors [9] play a determinant role. Among the latest ones, diet has been suggested to be looked up to: to date, many dietary patterns and their association with MS have been investigated [10,11], but data is still limited and inconclusive. In particular, previous observational studies found that patients with MS tend to have a less healthy diet or a more proinflammatory diet, compared with controls [12,13]. Emerging evidence suggests that adherence to the MedDiet might be associated with MS onset and progression [11,14,15,16], but studies are still scarce. In our previous publication [17] we reported that higher ultra-processed foods (UPFs) consumption was associated with moderate-to-high MS severity compared to lower consumption. Up to date, there are no data about the prevalence of MS in individuals with high consumption of UPFs but it is known that adhering to hyper-caloric diets high in animal-origin fats and sugars, like Western diets, can impact on the disease pathogenesis and course [18,19]. Recently, Alfredsson and colleagues [16] evaluated the risk of MS according to adherence to different dietary patterns (Western diet, MedDiet, and vegan/vegetarian diets). The authors reported that MedDiet was associated with lower risk of developing MS, compared with a Western-style diet, while no significant associations between vegetarian/vegan diet and low glycemic index diet MS risk were described. It has also been reported that a healthy eating pattern was associated with a 25% reduction in the risk of clinically isolated MS, in comparison to those who followed a Western-style regimen [20]. The detrimental effects of unhealthy dietary habits has been proved also in pediatric patients, with one study reporting a three-fold increased relapse risk each 10% increase in saturated fat intake [21]. 

Thus, available data, mainly derived from studies with an experimental design, suggest that the union of a balanced diet and an overall healthy lifestyle may determine an amelioration in several clinical parameters and in the quality of life for patients with MS (pwMS). Up to date, definitive and detailed recommendations concerning a specific dietary pattern for pwMS do not exist. The aim of this study is to investigate the possible difference in MS severity in a group of Italian pwMS according to their dietary habits, specifically their adherence to MedDiet. If the hypothesis of a link between MedDiet and MS is confirmed, our study would deeply investigate the possible effect of the whole dietary pattern and/or each component of MedDiet on MS risk in a population that lives in the Mediterranean region.

## 2. Materials and Methods

This is a single-center study, with a cross-sectional observational design. Patients were enrolled at IRCCS Foundation Mondino Neurological Institute between September 2020 and March 2022. For each participant, written informed consent was obtained after careful explanation of the study purposes and plan of action. PwMS were selected only if they met the following criteria: adult (age > 18 years) with a diagnosis of relapsing remitting MS or secondary progressive MS. Individuals with significant cognitive-cooperative deterioration, lack of cooperation, and/or PPMS were excluded. The study received ethical approval by the San Matteo Ethical Committee (P-20200064205, on 8 May 2020). The Declaration of Helsinki principles were followed when conducting this study. 

Data collection included demographics, nutritional and lifestyle information, adherence to MedDiet, smoking and physical activity evaluation and neurological assessment. As regards demographics, marital status was cataloged into the following groups: (i) unmarried and widowed, (ii) married. Three classes of educational level were created, as follows: (i) low (primary/secondary), (ii) medium (high school), and (iii) high (university).

A validated 110-items FFQ [22] and the Medi-Lite tool [23] were used to assess the patients’ dietary habits and level of adherence to MedDiet, respectively. Smoking habits were arranged as: (i) Non-smokers, (ii) smokers, and (iii) ex smokers. 

The International Physical Activity Questionnaire (IPAQ) was applied for the physical activity habits assessment. It is composed of five domains evaluating how much time in a week an individual spends being physically active. Patients were then classified into 3 groups on the basis of their physical activity level (low/moderate/high) [24]. The neurological disability of MS patients was evaluated through the Expanded Disability Status Scale (EDSS) [25] and the clinical impact of MS was determined using the Multiple Sclerosis Severity Score (MSSS) [26], taking into account the disease’s duration. Patients were classified in three groups according to their low, medium or high adherence to MedDiet. The MS classification is based on Herbert’s severity grading [27]. According to it, patients can be categorized into 6 classes of disability, roughly equal in distribution. MSSS values lower than 1.7 would correspond to a condition of mild MS; MSSS values between 1.7 and 3.4 would be defined moderate MS; MSSS grade from 3.4 to 5.0 would be compared to an intermediate MS severity; MSSS ranking between 5.0 and 6.7 would be considered accelerated MS; MSSS values from 6.7 to <8.3 would be consistent with advanced MS; and MSSS above 8.3 would be expressed as aggressive MS.

### 2.1. Food Frequency Questionnaire

Patients’ dietary habits were evaluated using a validated semi-quantitative FFQ, made up of 110 elements [22], of which the participant should have indicated the mean frequency of consumption during the previous half a year. Patients were individually contacted and telephonically interviewed by a skilled dietician (about 30–40 min). The interviewee had to ask how frequently on a monthly (“never”, “once” or “twice”), weekly (“once”, “two or three times”, “four or five times”) or daily (“once”, “two or three times”, “four or five times”) basis the participant habitually consumed each specific food. For most of the food included in the FFQ a specific portion size was indicated (i.e., fish 100 g). If the participant consumed a different quantity of food, the size was adjusted. Thus, the food frequencies reported by each pwMS for each item were transformed into daily intakes and their mean values were calculated (in g or ml). Energy, macro- and micro-nutrient intakes were computed referring to the standard food composition tables of the Italian Research Center for Foods and Nutrition [28]. Food seasonality was taken into account, asking the patient to report the food’s frequency of consumption during the span in which it was accessible and then it was harmonized to the whole year. 

### 2.2. Medi-Lite Score 

Adherence to MedDiet was evaluated through the Medi-Lite score [23], a literature-based quantitative score elaborated on the basis of data derived from cohort studies which evaluated the relationship between MedDiet adherence and health outcomes [29]. It considers the daily frequency and portion of consumption of 9 food groups: (1) fruit; (2) vegetables; (3) legumes; (4) cereals; (5) fish; (6) meat and cold cuts; (7) dairy products; (8) alcohol and (9) olive oil. The specific portion size is indicated and varies according to the single food group (i.e., 150 g for fruit, 70 g for legume). The participant has to choose which one of the three possible options (<1 portion a day, 1–1.1–1.5 portion a day and >2/2.5 portion a day) better suits his/her habitual consumption. The only exceptions are alcohol, for which the alcohol unit (1 alcohol unit = 12 g of alcohol) was used as a reference portion, and olive oil, whose frequency of consumption was classified in “occasionally”, “frequently” and “regularly”. Different scores were attributed to the food groups representative (the vegetable-origin foods, fish and oil) and those unrepresentative of the MedDiet (the animal-origin foods, including dairy products, and alcohol). As regards MedDiet food groups, 2 points were allocated when a higher consumption was reported, 1 point was given for the middle intake category and 0 points were assigned when a frequency of consumption lower than 1 portion a day was referred. In contrast, for food groups unrepresentative of the MedDiet, the lowest intake (less than 1 portion a day) received 2 points, 1 point was given when a moderate intake was reported and no points were assigned when the higher consumption was referred. The total score is calculated by summing the single points obtained for each food group and it can range from 0 to 18. The higher the score, the higher the adherence. To establish the three groups of adherence, the tertiles of MEDI-LITE score were calculated. 

### 2.3. EDSS and MSSS

The Expanded Disability Status Scale (EDSS) [25] is the most commonly applied clinical scale to quantify disability in MS, as it includes a multifunctional clinical assessment of all the systems usually evaluated in clinical practice. The global score is composed of the scores obtained evaluating each single functional system (Visual, Brainstem, Pyramidal, Cerebellar, Sensory, Cerebral, and sphincter).

We also calculated the Multiple Sclerosis Severity Score (MSSS) [26] that takes into account both the level of clinical disability, represented by EDSS, and the disease duration, therefore representing the clinical severity of the disease at any time during the disease course.

### 2.4. Statistical Analysis 

Percentages and frequencies of occurrence were used to indicate qualitative variables. The possible differences between the adherence groups (low vs. high) were evaluated through the Chi-squared test. Mean and standard deviations (SDs) were used for the quantitative variables. Student’s t-test was applied to investigate differences between groups of adherence. Logistic regression analysis was performed to study the association between levels of adherence to MedDiet and MS severity. Odds ratios (ORs) and 95% confidence intervals (CIs) were calculated both for an unadjusted model and a multivariate model adjusted for the possible confounding factors such as: energy intake (continuous, kcal/d), age (continuous, years), sex (male, female), BMI (normal, overweight, obese), marital status (unmarried/widowed, married), educational status (low, medium, high), smoking status (no smokers, smokers and ex smokers), and physical activity level (low, moderate, high). All reported P-values were based on two-sided tests and compared to a significance level of 5%. The statistical analysis was performed through SPSS 21 (SPSS Inc., Chicago, IL, USA) software.

## 3. Results

In this study 130 pwMS were enrolled but 24 were excluded from the analysis for the following reasons: missing data (*n* = 15), refusal to complete the telephonic interview (*n* = 5) and disease type (*n* = 4). The final sample was composed of 106 participants. Low, medium and high Medi-Lite score groups were constituted of 40, 38 and 28 patients, respectively. As detailed in Table 1, no significant differences were found in the background characteristics between the groups. 

Table 2 represents the daily intakes of the main food groups of MedDiet. As expected, participants in the higher adherence group had significantly higher consumption of cereals (266.2 ± 73.8 g/d vs. 148.1± 109.6 g/d) and whole grains (78.9 ± 90.4 g/d vs. 24.4 ± 48.6 g/d)), fruit (590.9 ± 204.5 vs. 249.3 ± 188.9 g/d) and vegetables (338.8 ± 196.3 vs. 135.0 ± 75 g/d), legumes (50.3 ± 42.4 vs. 24.3 ± 35.6 g/d), fish (64.7 ± 32.4 vs. 35.5 ± 30.1 g/d) and olive oil (8.4 ± 2.4 vs. 6.6 ± 3.5 g/d). 

As shown in Table 3, those with higher Medi-Lite score had also a greater intake of energy, macronutrients (carbohydrates, protein, total fat, MUFA, PUFA), fiber, and micronutrients (except for vitamin D, vitamin B12 and pyridoxine) compared to the other groups. 

Considering the regression analysis reported in Table 4, higher adherence to MedDiet, expressed by the Medi-Lite score, was associated with a higher probability of having a mild-to-moderate MS compared to low-medium adherence in both models (OR = 5.28, 95% CI: 1.58, 17.69 and OR = 6.18, 95% CI: 1.44, 26.59, respectively). Model 1 also showed nearly 3-fold likelihood in those with medium adherence to MedDiet (OR = 2.83, 95% CI: 1.03, 7.75) but significance was lost when adjusting for the other confounding factors (Model 2). 

When studying the single constituent of the Medi-Lite score, none of them was individually associated with MS severity (Table 5).

## 4. Discussion

The objective of the present study was to examine the relation between level of adherence to the MedDiet and MS severity in a sample of 106 patients. The findings show that higher adherence to the MedDiet was associated with lower MS severity compared to those reporting poorer adherence. Importantly, no individual constituent of the Mediterranean diet was independently associated with MS severity, suggesting the synergistic role of the overall dietary pattern over individual food intake. PwMS with higher MedDiet adherence consumed on average 4–5 portions of cereals, including 1 portion of unrefined grains, 2 portions of vegetables, 3 portions of fruit and about 1 portion of olive oil. Moreover, their consumption of fish and legumes was doubled compared to the other groups. These notable differences in food intakes can explain the important differences in energy, macronutrients and micronutrients that we found. Although some studies [30,31] reported that high calories intakes are not desirable in pwMS, the specific individual needs and the food sources from which energy derives should be taken into account. In our study, the greater energy intake was linked to a higher consumption of vegetable origin foods and fish and did not derive from animal-origin foods. These choices were reflected also analyzing the different macro and micronutrients intakes. In fact, the increase in total fat intake was mediated by the rise of monounsaturated and polyunsaturated fatty acids, while no differences were found in the saturated fatty acids. As well, carbohydrates higher consumption matched with a greater intake of fiber, derived from fruit, vegetables, legumes and whole grains.

Our findings corroborates the hypothesis that the possible beneficial effect of MedDiet could be mediated by the whole dietary pattern, and is not linked to a single food group or constituent of this regimen. Surely, the entire lifestyle is important: physical activity [32,33] and smoking habits [34,35] impact on MS progression but, when adjusting for these and all the possible confounding factors, the association between high adherence to MedDiet and MS severity was not modified

Although not entirely comparable due to different tools used to evaluate both exposure to the Mediterranean diet and outcome measures, the findings of this study agree with previously published research-based data. A multicenter study conducted in Italy on 435 consecutive pwMS showed that MedDiet adherence scores were inversely correlated with EDSS and MSSS, but not correlated with the total number of relapses [3]. Also a study conducted on 563 patients recruited in a US center reported that the MedDiet adherence score was linearly associated with the MS Functional Composite, which is used to assess deambulation, arm/hand function, and cognitive function. In this study MedDiet adherence was also related to patient-reported outcomes such as: disability, ambulatory dysfunction, depression, anxiety and fatigue as well as attenuated the consequences of disease progression and duration of disability. However, a study involving 102 Turkish MS patients (mean age of 33 y) found no relation with MS-related symptoms but a linear inverse association with fatigue severity scale was reported. Among the specific dietary components investigated, the authors concluded that reduced intakes of red meat, saturated fatty acids, and sweets, as well as higher fish consumption, might have an impact on MS symptoms or fatigue severity [36]. 

Another study conducted in Turkey on 95 pwMS showed an association between MedDiet adherence, EDSS and physical and mental quality of life parameters (CPH and CMH), independently of the disease course [37]. Furthermore, a study conducted in Iran on 261 MS patients (mean age of 38.9 y) reported no association between following a Mediterranean-type regimen and EDSS, yet significantly related to higher Physical and Mental Health Composite Scores [38]. In contrast, another Iranian study conducted on 478 pwMS (mean age of 38 y) reported no significant relationship between MedDiet adherence and severity of symptoms [39]. 

Also few intervention studies have been conducted to test whether the adoption of a MedDiet would affect MS symptoms severity. A network meta-analysis of trials conducted in the USA up to 2021 showed that intervention studies using the MedDiet demonstrated significant effects on fatigue symptoms compared to controls [40]. A more recent intervention pilot study proposing a Mediterranean dietary program to 36 Iranian pwMS followed-up for 6 months showed a statistically significant decline in the trajectory of Neurological Fatigue Index-MS scores, a trend toward reduced MS Impact Scale-29 scores, and a decrease in EDSS over the years in comparison to the control group [41]. Another trial conducted on 180 MS individuals for 6 months showed that a modified MedDiet would reduce the Modified Fatigue Impact Scale, although no considerable improvement for EDSS was detected [42].

Several mechanisms related to the main features of the MedDiet have been hypothesized to explain the possible benefits of adhering to this dietary regimen concerning the MS symptoms severity [43]. The main rationale includes the presence of various compounds and their relative food sources hypothesized to be beneficial for brain health [44]. The MedDiet is primarily a plant-based dietary pattern characterized by abundance in fruit and vegetables, which are naturally rich in antioxidants, such as vitamins and (poly)phenols. Certain vitamins, including vitamin D, vitamin E, and vitamin C have been shown in experimental models to promote neural stem cell proliferation and differentiation into oligodendrocytes, which carry out remyelination [45], decrease tau-protein hyperphosphorylation and increase superoxide dismutase and brain-derived neurotrophic factor levels [46], and more broadly to potentially prevent inflammation and subsequent oxidative damage [47]. Other non-nutrient compounds, such as (poly)phenols, have been demonstrated to exert neuroprotection through directly counteracting impaired neurogenesis, mitochondrial dysfunction, microglia activation and neuroinflammation (those compounds passing through the blood-brain barrier) [48] or modulating the gut microbiota toward an anti-inflammatory profile and immunomodulatory activity ultimately resulting, among others, in neuroprotective effects [49]. The common inclusion of fish (rich in EPA and DHA) and plant-derived ω-3 fatty acids (such as ALA that are naturally present in nuts and seeds) has been pointed out as potentially beneficial in individuals with neurodegenerative disorders due to their anti-inflammatory and neurotrophic effects in the central nervous system [50]. In fact, ω-3 fatty acids have been demonstrated to increase fluidity and function of neuron membranes by improving signal transduction through regulation of membrane G-proteins, Na/K-dependent ATPase enzymes, and protein kinase C [51]. Moreover, their involvement in inflammatory processes may counteract neuroinflammation, reduce the production of pro-inflammatory cytokines and increase the synthesis of pro-resolving lipid mediators and neurotrophic factors (i.e., brain-derived neurotrophic factor) [52]. Conversely, higher adherence to the MedDiet has been related with lower intake of unhealthy and ultra-processed foods (UPFs) [53,54], whose high consumption may be related to detrimental effects on the gut microbiota, establishment of dysbiosis, and consequent alteration in the immune system including long-distance effects on brain health [55]. In line with such a hypothesis, we previously reported that higher intakes of UPFs was associated with worse MS severity [17], but further studies are needed to corroborate our findings. A graphical synthesis of the mechanisms through which vitamins, bioactive compounds and UPFs can impact on multiple sclerosis is presented in Figure 1.

The present study has some limitations. First, the observational design can only allow us to point out the association between MedDiet adherence and MS severity, without deepening the causal-effect mechanisms. Second, the sample size is relatively small (yet in line with other existing previously cited studies) and belongs to a specific geographical area, thus not permitting the generalization of the results. Third, the use of the FFQ allowed us to collect dietary data from a longer time period, better representing the eating habits of the participants, the estimation of dietary intakes might lead to untrue reporting (which could have been underestimated or overestimated) due to recall bias or social desirability. Moreover, the use of the telephonic interview reduced the risk of missing data, although it could have influenced the patients’ responses. Finally, the possible presence of comorbidities, which could have influenced the patients’ dietary habits, was not explored.

## 5. Conclusions

To conclude, higher adherence to the MedDiet was associated with lower MS severity. It remains unclear whether the effects of specific dietary components included in the MedDiet may impact the health status at disease onset or can slow down the symptoms due course of disease. Future studies are needed to reproduce our findings and should focus on answering the latter raised question.

## Figures and Tables

**Figure 1 nutrients-15-04009-f001:**
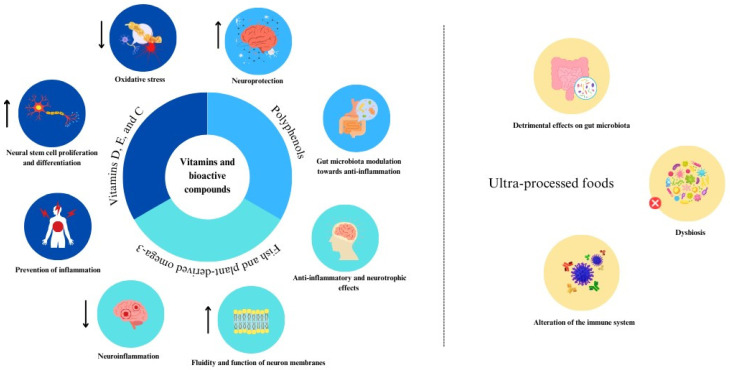
Graphical synthesis of the mechanisms through which vitamins, bioactive compounds and ultra-processed foods can impact on multiple sclerosis. Down arrows symbolize a reduction effect on the specific mechanisms, while up arrows indicate an increase in the pathway.

**Table 1 nutrients-15-04009-t001:** Participants’ background characteristics based on the tertiles of Medi-Lite score (*n* = 106).

		Medi-Lite Score		
	Low (*n* = 40)	Medium (*n* = 38)	High (*n* = 28)	*p*-Value
Age, mean (SD)	49.2 (13.4)	48.1 (10.3)	52.2 (11.9)	0.382
*Sex, n (%)*				0.740
Men	16 (40.0)	12 (31.6)	10 (35.7)	
Women	24 (60.0)	26 (68.4)	18 (64.3)	
*Smoking status, n (%)*				0.325
Non-smokers	19 (47.5)	21 (55.3)	20 (71.4)	
Smokers	10 (25.0)	7 (18.4)	5 (17.9)	
Ex smokers	11 (27.5)	10 (26.3)	3 (10.7)	
*Educational level, n (%*)				0.175
Low	10 (25.0)	3 (7.9)	6 (21.4)	
Moderate	21 (52.5)	20 (52.6)	11 (39.3)	
High	9 (22.5)	15 (39.5)	11 (39.3)	
*Marital status, n (%)*				0.426
Unmarried/widowed	17 (42.5)	12 (31.6)	13 (46.4)	
Married	23 (57.5)	26 (68.4)	15 (53.6)	
*Physical activity level, n (%)*				0.235
Low	17 (42.5)	19 (50.0)	9 (32.1)	
Moderate	17 (42.5)	12 (31.6)	9 (32.1)	
High	6 (15.0)	7 (18.4)	10 (35.7)	
*BMI status, n (%)*				0.485
Normal	29 (72.5)	27 (71.1)	15 (53.6)	
Overweight	10 (25.0)	9 (23.7)	11 (39.3)	
Obese	1 (2.5)	2 (5.3)	2 (7.1)	
*Current therapy, n (%)*				0.389
No	16 (40.0)	10 (26.3)	8 (28.6)	
Yes	24 (60.0)	28 (73.7)	20 (71.4)	
*Therapy type, n (%)*				0.571
First line	17 (70.8)	22 (78.6)	16 (84.2)	
High potency	7 (29.2)	6 (21.4)	3 (15.8)	

**Table 2 nutrients-15-04009-t002:** Daily intakes of the major food groups based on the Medi-Lite score, expressed as mean and standard deviation.

		Medi-Lite Score		
	Low	Medium	High	*p*-Value
Cereals (g/d)	148.1 (109.6)	190.0 (105.5)	266.2 (73.8)	*<0.001*
Whole grains (g/d)	24.4 (48.6)	63.3 (81.9)	78.9 (90.4)	*0.008*
Vegetables (g/d)	135.0 (75.0)	213.8 (93.7)	338.8 (196.3)	*<0.001*
Fruit (g/d)	249.3 (188.9)	329.8 (226.1)	590.9 (204.5)	*<0.001*
Legumes (g/d)	24.3 (35.6)	46.8 (51.2)	50.3 (42.4)	*0.024*
Nuts (g/d)	18.0 (27.7)	15.4 (18.9)	23.3 (40.6)	0.556
Fish (g/d)	35.5 (30.1)	46.5 (37.1)	64.7 (32.4)	*0.003*
Eggs (g/d)	1.1 (1.0)	1.6 (1.4)	1.2 (0.9)	0.126
Red meat (g/d)	15.5 (17.6)	17.3 (12.9)	13.5 (10.6)	0.572
Processed meat (g/d)	12.8 (13.3)	11.4 (10.9)	10.3 (12.3)	0.690
Dairy products (g/d)	225.4 (193.4)	195.1 (151.9)	221.9 (150.4)	0.698
Alcohol (g/d)	7.4 (15.8)	0.7 (1.3)	6.8 (8.2)	0.555
Olive oil (g/d)	6.6 (3.5)	8.9 (2.1)	8.4 (2.4)	*0.001*

d = day. In italics are highlighted the statistically significant differences.

**Table 3 nutrients-15-04009-t003:** Macronutrients and micronutrients daily intakes according to the Medi-Lite score, expressed as mean and standard deviation.

		Medi-Lite Score		
	Low	Medium	High	*p*-Value
Energy intake (kcal/d)	1596.0 (489.3)	2005.1 (602.2)	2322.5 (496.2)	*<0.001*
Energy intake (kJ/d)	6488.3 (2076.3)	8123.3 (2569.7)	9458.7 (2051.0)	*<0.001*
Protein (g/d)	63.5 (20.7)	80.6 (28.2)	94.2 (22.5)	*<0.001*
Fat (g/d)	55.1 (21.2)	68.6 (26.5)	67.9 (24.8)	*0.027*
Cholesterol (mg/d)	150.5 (83.7)	169.9 (95.9)	168.5 (64.5)	0.535
Saturated fatty acids (%)	22.2 (10.7)	26.5 (14.3)	25.8 (10.6)	0.256
MUFA (%)	22.8 (8.3)	28.7 (9.3)	28.5 (9.4)	*0.007*
PUFA (%)	9.2 (4.1)	11.9 (4.9)	12.7 (5.4)	*0.006*
Carbohydrates (g/d)	219.0 (80.8)	276.5 (84.5)	350.1 (73.0)	*<0.001*
Total fiber (g/d)	20.1 (8.4)	30.8 (10.5)	41.2 (13.4)	*<0.001*
Vitamin A retinol eq (μg/d)	642.3 (247.3)	870.8 (264.3)	1230.2 (520.6)	*<0.001*
Vitamin C (mg/d)	115.3 (49.1)	167.3 (89.8)	238.6 (97.1)	*<0.001*
Vitamin E (mg/d)	6.8 (2.7)	9.1 (2.8)	10.9 (3.8)	*<0.001*
Vitamin D (μg/d)	3.2 (2.8)	4.2 (3.5)	5.1 (2.9)	0.053
Vitamin B12 (μg/d)	4.4 (2.2)	5.2 (2.9)	5.9 (3.0)	0.067
Thiamin (mg/d)	1.3 (0.6)	1.8 (0.6)	2.2 (1.0)	*<0.001*
Riboflavin (mg/d)	1.7 (0.8)	2.1 (0.8)	2.7 (1.0)	*<0.001*
Niacin (mg/d)	16.3 (4.9)	21.3 (5.9)	26.5 (7.0)	*<0.001*
Pyridoxine (mg/d)	1277.6 (393.0)	1478.4 (418.2)	1456.2 (449.1)	0.076
Vitamin B9 (μg/d)	264.9 (84.7)	388.2 (126.3)	506.2 (184.0)	*<0.001*
Sodium (mg/d)	1872.9 (782.7)	2303.2 (832.0)	2457.9 (541.7)	*0.004*
Potassium (mg/d)	2644.3 (798.6)	3517.7 (994.7)	4632.7 (1373.6)	*<0.001*
Iron (mg/d)	10.3 (3.6)	15.0 (4.6)	18.7 (5.6)	*<0.001*
Calcium (mg/d)	698.8 (341.1)	841.6 (496.2)	989.5 (326.5)	*0.015*
Phosphorus (mg/d)	1070.9 (374.2)	1365.7 (486.8)	1634.0 (465.5)	*<0.001*
Magnesium (mg/d)	296.6 (97.0)	407.0 (118.4)	515.6 (148.8)	*<0.001*
Zinc (mg/d)	8.5 (2.7)	11.2 (3.8)	13.9 (3.8)	*<0.001*
Copper (mg/d)	1.5 (0.5)	2.1 (0.7)	2.6 (0.9)	*<0.001*
Selenium (μg/d)	74.6 (35.8)	100.4 (39.2)	121.2 (25.7)	*<0.001*

d = day. In italics are highlighted the statistically significant differences.

**Table 4 nutrients-15-04009-t004:** Association between the Mediterranean diet adherence score and mild-to-moderate MS (in comparison to low adherence), expressed as odds ratio and the respective 95% confidence interval.

	Mild-to-Moderate Multiple Sclerosis, OR (95% C.I.)		
	Low	Medium	High
Mediterranean diet adherence			
Model 1 *	1	2.83 (1.03, 7.75)	5.28 (1.58, 17.69)
Model 2 **	1	3.12 (0.97, 10.01)	6.18 (1.44, 26.59)

* Model 1 was adjusted for energy intake (kcal). ** Model 2 was additionally adjusted for BMI (normal, overweight, obese), sex (male/female), educational level (low, medium, high), smoking habits (no smokers, smokers, ex smokers), age (continuous, years), physical activity (low, moderate, high), marital status (unmarried/widowed, married).

**Table 5 nutrients-15-04009-t005:** Association between the single constituents of the Mediterranean diet adherence score and mild-to-moderate MS, expressed as odds ratio and 95% CIs.

	Mild-to-Moderate Multiple Sclerosis, OR (95% C.I.)
Mediterranean diet components	
Fruit (>1 serv/d, 150 g/d)	0.95 (0.18, 4.92)
Vegetable (>1 serv/d, 100 g/d)	0.38 (0.11, 1.39)
Legume (>1 serv/w, 70 g/w)	1.98 (0.69, 5.76)
Cereals (>1 serv/d, 130 g/d)	1.30 (0.53, 3.14)
Fish (>1 serv/w, 100 g/w)	1.36 (0.28, 6.66)
Meat (<1.5 serv/d, 120 g/d)	1.30 (0.38, 4.41)
Dairy products (<1 serv/d, 180 g/d)	0.47 (0.20, 1.11)
Alcohol (1–2 AU/d, 12 g/d)	0.51 (0.29, 9.31)
Olive oil (>1 portion/d, 5 g/d)	2.05 (0.35, 11.72)

AU, alcohol unit; d = day; w = week.

## Data Availability

The data generated during the current study are available from the corresponding author upon reasonable request.

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
