# Peer review of "Adherence to Mediterranean Diet Is Associated with Multiple Sclerosis Severity"

_nutrients, 2023, doi:10.3390/nu15184009_

Round 1
Reviewer 1 Report
Minor comments: The introduction is very condensed, and it would be best to include background information on the prevalence of multiple sclerosis associated with diets based on ultra-processed foods. - In the abstract, the acronym MSSS needs to be included. - Line 150 is misspelled the acronym (SM). - It is recommended in the table to highlight the P values that represent statistically significant differences. - Line 139 correct the word "sphincter".
In the discussion, it is necessary to highlight the importance of the findings of the work rather than compare it if they agree or not with other published works. - Table 5. To check that the units are in the same format for all the variables.
Author Response
We sincerly thank you reviewer 1 for the helpful comments that allowed us to improve the manuscript. Below, our responses point by point.
Minor comments: The introduction is very condensed, and it would be best to include background information on the prevalence of multiple sclerosis associated with diets based on ultra-processed foods.
Thank you for the comment, unfortunately there is no data (excluding our previous study) on the prevalence of multiple sclerosis associated with ultra-processed foods consumption. We reported data on the relationship between MS and Western diets (which are characterized by the consumption of food products with nutritional features similar to ultra-processed foods) (lines 55-67)
- In the abstract, the acronym MSSS needs to be included.
Thank you for the suggestions, we inserted the acronym in the abstract (lines 22-23)
- Line 150 is misspelled the acronym (SM).
Thank you, it probably was a typing error. We corrected it
- It is recommended in the table to highlight the P values that represent statistically significant differences.
We thank you for the advice, we put in italics the statistically significant differences
- Line 139 correct the word "sphincter".
Thank you, we corrected it
- In the discussion, it is necessary to highlight the importance of the findings of the work rather than compare it if they agree or not with other published works.
Thank you for the comment, we added a paragraph about it (lines 250-255)
- Table 5. To check that the units are in the same format for all the variables.
Thank you, we have checked and it is correct that some items are expressed as daily intakes and others as weekly. In fact, the MediLite score uses these frequencies of consumption, which are constructed in order to reflect the ones recommended in the Mediterranean pattern, as described by Bach Faig et al. However, we added the footnotes for explanation.
Kind regards,
Monica Guglielmetti and co-authors

Reviewer 2 Report
Dear Authors,
here below i present my detailed comments to your interesting work:
- please, enrich the introduction in a more precise statement on what was novel in your study and what has been already proved by other authors to direct a reader into new findings.
- please, add a figure with the data presented in the discussion part of the manuscript that are related to the actual proved bioactivities of the single components of the Mediterranaean diet that have impact on the status of the brain
- i have been wondering abou tthe tested group of people. as it is stated in the manuscript, this is a population from one area. Please, enrich your discussion and develop in a few sentences the impact of the area that the people are living in. How propable is that some of the representatives of the tested group were brought up already consuming this type of diet. Maybe it could have impact on how they react in this study. How about other nationalities that were not used to eating Mediterranaen style? are there any trials that could be cited here? are the latter groups more prone to the effects of the MedDiet?
- have the Authors gothered any information about the length of sleep and style of life? please, comment on that
Author Response
We sincerely thank reviewer 2 for the precious comments, that allowed us to improve the manuscript. Below, our comments point by point
- please, enrich the introduction in a more precise statement on what was novel in your study and what has been already proved by other authors to direct a reader into new findings.
Thank you for the suggestion, we expanded the introduction and added more precise statements on what our study wanted to introduce as a novelty in the field. (lines 49-79)
- please, add a figure with the data presented in the discussion part of the manuscript that are related to the actual proved bioactivities of the single components of the Mediterranean diet that have impact on the status of the brain
Thank you, we added figure 1 at the end of the text
- I have been wondering about the tested group of people. as it is stated in the manuscript, this is a population from one area. Please, enrich your discussion and develop in a few sentences the impact of the area that the people are living in. How probable is that some of the representatives of the tested group were brought up already consuming this type of diet. Maybe it could have an impact on how they react in this study. How about other nationalities that were not used to eating Mediterranean style? Are there any trials that could be cited here? Are the latter groups more prone to the effects of the MedDiet?
Thank you for pointing out these interesting elements. We will try to comment by points:
- How probable is that some of the representatives of the tested group were brought up already consuming this type of diet. Maybe it could have an impact on how they react in this study.
A: This is an observational study and not an interventional study. We evaluated the dietary habits of pwMS, some were adherent to MedDiet and others were not. The distribution of patients in the three groups of adherence was quite equipollent, so there was no effect on the representativeness of the sample.
- How about other nationalities that were not used to eating Mediterranean style? Are there any trials that could be cited here? Are the latter groups more prone to the effects of the MedDiet?
A: In the discussion we included studies from different countries (USA, Iran, Turkey,...) and, excluding one study, all the others supported the beneficial effect of MedDiet. It is not possible to compare the "effectiveness" of the diet among the studies (and so the different populations) because of different outcomes and methods used, so we are not able to answer your latter interesting question.
- have the Authors gathered any information about the length of sleep and style of life? please, comment on that
Unfortunately we do not have any information about the sleep duration and quality but thank you for the suggestion, we surely will study this aspect in future. As regards the other lifestyle factors, we collected information about smoking habits and physical activity, as reported in the manuscript

Round 2
Reviewer 2 Report
Dear Authors,
thank you for introducing the requested changes. I have no more comments to your work.